# Exploiting Emotion-Semantic Correlations for Empathetic Response Generation

**Zhou Yang**[1,2], **Zhaochun Ren**[3], **Yufeng Wang**[1,2], **Xiaofei Zhu**[4], **Zhihao Chen**[1,2],
**Tiecheng Cai**[1,2], **Yunbing Wu**[1,2], **Yisong Su**[1,2], **Sibo Ju**[1,2], **Xiangwen Liao**[1,2*]

[1]College of Computer and Data Science, Fuzhou University; [2]Digital Fujian Institute of Financial Big Data, Fuzhou, China

[3]Leiden University, Leiden, The Netherlands

[4]College of Computer Science and Technology, Chongqing University of Technology, Chongqing, China

`{200310007, 211027083, n180320046, 210310002, 221010003, 221010003}@fzu.edu.cn`

`z.ren@liacs.leidenuniv.nl zxf@cqut.edu.cn {wyb5820, liaoxw}@fzu.edu.cn`

## Abstract

Empathetic response generation aims to generate empathetic responses by understanding the speaker's emotional feelings from the language of dialogue. Recent methods capture emotional words in the language of communicators and construct them as static vectors to perceive nuanced emotions. However, linguistic research has shown that emotional words in language are dynamic and have correlations with other grammar semantic roles, i.e., words with semantic meanings, in grammar. Previous methods overlook these two characteristics, which easily lead to misunderstandings of emotions and neglect of key semantics.

To address this issue, we propose a dynamical Emotion-Semantic Correlation Model (ESCM) for empathetic dialogue generation tasks. ESCM constructs dynamic emotion-semantic vectors through the interaction of context and emotions. We introduce dependency trees to reflect the correlations between emotions and semantics. Based on dynamic emotion-semantic vectors and dependency trees, we propose a dynamic correlation graph convolutional network to guide the model in learning context meanings in dialogue and generating empathetic responses. Experimental results on the EMPATHETIC-DIALOGUES dataset show that ESCM understands semantics and emotions more accurately and expresses fluent and informative empathetic responses. Our analysis results also indicate that the correlations between emotions and semantics are frequently used in dialogues, which is of great significance for empathetic perception and expression.[1]

## 1 Introduction

Attracting an increasing amount of attention, empathetic response generation aims to generate empa-

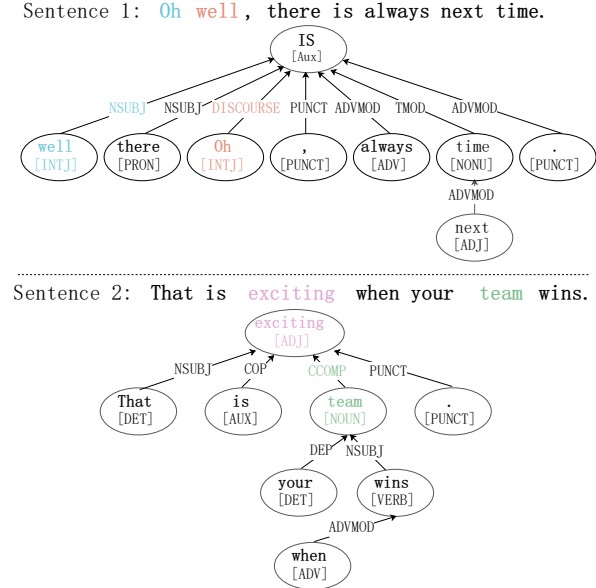

Figure 1: Examples from the EMPATHETIC-DIALOGUES dataset. Sentence 1 shows the variability of emotional words. Sentence 2 shows the correlations of emotional words with semantic roles.

thetic responses by perceiving the speaker's emotional feelings (Rashkin et al., 2019; Zhong et al., 2021, 2020; Liang et al., 2021; Zheng et al., 2021; Liu et al., 2021; Wang et al., 2021).

Early methods perceive the speaker's feelings by understanding the holistically semantics and emotions expressed in the language of the context (Rashkin et al., 2019; Lin et al., 2019; Majumder et al., 2020). These methods are prone to generate trivial and uninformed responses, ascribed to the neglect of nuances of human emotion in dialogues (Li et al., 2020). To address this issue, recent methods detect emotional words in the language of the communicators and build them as static vectors to perceive subtle emotions (Li et al., 2020; Sabour et al., 2022; Kim et al., 2021; Gao et al., 2021; Li et al., 2022; Kim et al., 2022).

However, according to linguistic re-

---

*Corresponding author.

[1]Our code is available at `https://github.com/zhouzhouyang520/EmpatheticDialogueGeneration_ESCM`

search (Foolen et al., 2012; Dirven, 1997; Osmond, 1997; Radden, 1998), focusing on emotional words while ignoring their characteristics in the expression process leads to emotional misunderstandings and the neglect of words with important semantic information. As an important theory of emotional expression in linguistics, the conceptualization of emotions (Foolen et al., 2012) suggests that emotional words have two important characteristics in the expression process: variability and relevance. Variability is that the affection of emotional words changes dynamically during the expression process. For example, "well" generally carries a positive meaning, but in sentence 1 (shown in Figure 1), it is used as an interjection to express a neutral emotion. Using static vectors (such as Embedding (Pennington et al., 2014; Mikolov et al., 2013) or VAD (Mohammad, 2018)) to represent this dynamic emotion, previous methods are prone to misunderstand this sentence as positive.

Relevance refers to the grammatical correlations between emotional words and words carrying semantic meaning, which plays an important role in understanding emotions and semantics. For example, as shown in Figure 1, sentence 2 expresses the "exciting" emotion due to the victory of "team". "Team" is the primary subject described in the sentence, carrying key semantic information. Through the "[ADJ]-CCOMP-[NOUN]" correlation, the emotional word "exciting" directly modifies "team". Compared to previous work that did not consider such correlations, the model is more likely to identify key semantic words that are directly associated with emotional words through these types of syntax-meaningful relationships. Therefore, focusing on the variability and relevance of emotional words can promote the correct recognition of emotions and the detection of important semantics.

Therefore, we propose a dynamical Emotion-Semantic Correlation Model (ESCM) for empathetic dialogue generation. ESCM dynamically constructs emotion-semantic vectors through the interaction of context and emotions. By encoding emotion-semantic vectors, the model dynamically adjusts emotions and semantics in the context to capture the variability of emotional words. To reflect the correlations between emotions and semantics clearly, we introduce a dependency tree. Based on the dynamic emotion-semantic represen-

tation and the dependency tree, ESCM proposes a dynamic correlation graph convolutional network to guide the model to capture the correlations between emotions and semantics clearly. By learning dynamic emotion-semantic representations and their correlations, ESCM accurately understands the emotions of the dialogue and captures important semantics to generate more empathetic responses.

We conduct experiments on the EMPATHETIC-DIALOGUES dataset (Rashkin et al., 2019). The results show that the ESCM model accurately understands the dialogue and generates grammatically fluent and informative empathetic responses. Furthermore, we extract and statistically analyze the common correlation structures in dialogues from the Empathetic-Dialogue dataset. The results indicate that the correlations between emotion and semantics are frequently and extensively utilized in expressing emotions during conversations. Additionally, the results of our analysis of correlation structures are consistent with linguistic conclusions (Foolen et al., 2012).

To sum up, our contributions are as follows:

- We introduce the expressive characteristics of emotions in linguistics, including the variability of emotions and the correlations between emotions and semantics, to enhance the understanding of the meaning in conversations.

- We propose the ESCM model, which constructs dynamic emotion-semantic vectors to adjust the dynamics of emotions, and leverages a dependency tree-based dynamic correlation graph convolutional network to learn correlations, in order to generate empathetic responses.

- Experiments on the EMPATHETIC-DIALOGUE dataset demonstrate the effectiveness of ESCM. Furthermore, additional statistical and analytical experiments show that the correlations in dialogue are consistent with psychological research.

## 2 Related Work

Empathetic response generation refers to empathetically responding by perceiving emotional feelings in the language of the speaker (Rashkin et al., 2019).

Early approaches explore the overall emotions of the conversation. Rashkin et al. (2019) intro-

duce emotion representation generated by a pretrained emotion classifier to learn and express specific types of emotions in the conversation. However, emotions expressed in responses are often diverse rather than specific (Lin et al., 2019). Therefore, Lin et al. (2019) utilize multiple professional emotion listeners to express various appropriate emotions. Majumder et al. (2020) group multiple conversation emotions by polarity and simulate the speaker's emotions to generate empathetic responses.

These methods focus on the overall emotions of the conversation and ignore nuanced emotions (Li et al., 2020). To capture nuanced emotions, Li et al. (2020) extract emotional words through the NRC Emotion Lexicons (Mohammad and Turney, 2013) and integrate them into the model. Gao et al. (2021) and Kim et al. (2021) introduce emotional cause detection models to capture emotional words and perceive nuanced emotions. Li et al. (2022) enhance emotional representation in the context with additional knowledge, which helps detect emotional words. Sabour et al. 2022 use commonsense reasoning knowledge to infer nuanced emotions in the conversation. Kim et al. 2022 employ pre-trained models to detect word-level emotion and keywords to detect the nuanced emotion in dialogues. These methods detect emotional words with nuanced emotions in the language of the conversation and use static vectors such as word embeddings or VAD to represent emotional words.

Overall, early approaches ignore emotional words with nuanced emotions. Recent methods ignore two major characteristics of emotional words in linguistic expression: variability and correlation. Unlike these methods, we consider the two characteristics of emotional words and propose a dynamic emotion-semantic correlation model to better understand the conversation.

# 3 Method

## 3.1 Task Formulation

Given a dialogue context $D = [U_1, U_2, ..., U_M]$ of two interlocutors, our model needs to accurately perceive the emotions and semantics in the dialogue context and generate empathetic responses $Y = [y_1, y_2, ..., y_j, y_N]$. Here, $U_i = [w_1^i, w_2^i, ..., w_{m_i}^i]$ represents the i-th utterance containing $m_i$ words. $Y$ is a response containing N words.

## 3.2 Overview

We propose ESCM, which takes into account the dynamic emotions and semantics in the dialogue and their correlations. The proposed model is a transformer-based model with encoder-decoder architecture. To accurately perceive the content of the dialogue, we mainly reconstruct the encoder. As shown in Figure 2, ESCM mainly consists of three parts: (1) a context encoder (**Section 3.3**), which is a standard encoder structure and is used to understand the semantics of the dialogue; (2) a dynamic correlation encoding module (**Section 3.4**), which includes the construction of dynamic correlation vectors and the encoding of a dynamic correlation graph convolutional network. It learns the correlations between emotions and semantics; (3) emotion and response predicting module (**Section 3.5**), which completes the functions of emotion prediction and response generation.

## 3.3 Context Encoder

As with previous methods (Li et al., 2020, 2022; Sabour et al., 2022), we concatenate the utterances of the dialogue context and prepend $[CLS]$ as the whole sequence token to form the context input $C = = [CLS] \oplus U_1 \oplus U_2 \oplus ... \oplus U_M$. Here, $\oplus$ denotes the concatenation symbol. To input the context $C$ into the model, we convert $C$ into context word embeddings $E_c$. Then, we sum up the word embeddings $E_c$, position embeddings, and state embeddings to form the semantic embeddings $\widetilde{E}_c$. The state embeddings are used to distinguish between speaker or responder types and are randomly initialized. To understand the semantics of the dialogue, we feed the semantic embeddings $\widetilde{E}_c$ into the context encoder $Enc_{ctx}$ to obtain the context semantic representation $H_k$:

$$H_{ctx} = Enc_{ctx}(\widetilde{E}_c) \tag{1}$$

where $H_{ctx} \in R^{L \times d}$, L is the length of the context sequence, and d represents the hidden size of the encoder.

## 3.4 Dynamic Correlation Encoding Module

Dynamic correlation encoding consists of two submodules: (1) Dynamic emotion-semantic vectors. It brings the model the ability to flexibly adjust emotions and semantics, making the representation of context more reasonable. (2) Dynamic correlation graph convolutional network. This module

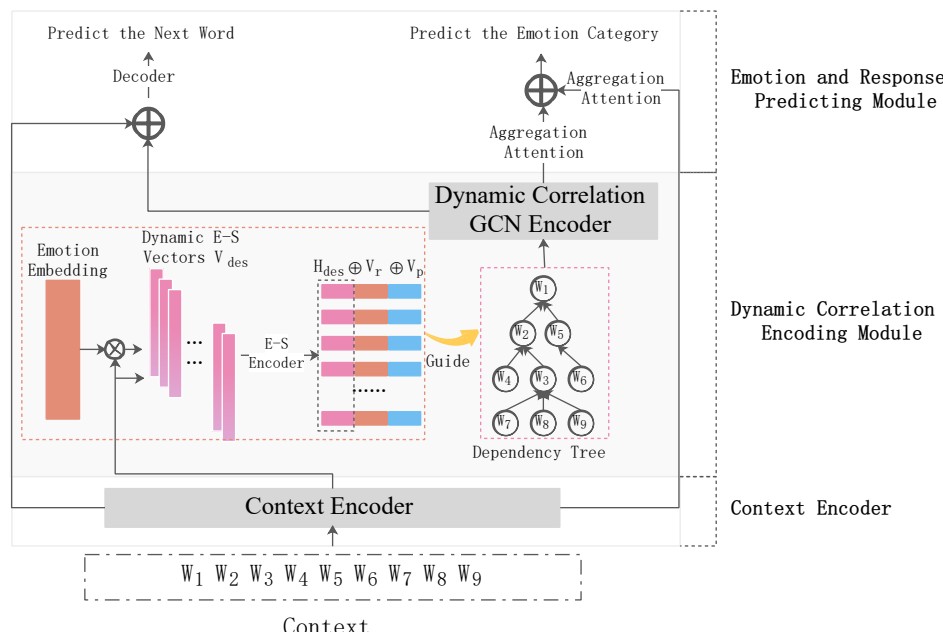

Figure 2: An overview of ESCM. ESCM consists of three main key modules: (1) a context encoder (**Section 3.3**), which encodes the semantic of context. (2) a dynamic correlation encoding module (**Section 3.4**), which learns the correlations between emotions and semantics. (3) emotion and response predicting module (**Section 3.5**), which predicts dialog emotion categories and generates empathetic responses.

uses emotions, semantics, part-of-speech, and dependency types to guide the model to discover and aggregate words with strong correlations, in order to more accurately understand the emotions and semantics of the conversation.

**Dynamic Emotion-Semantic Vectors**. Context has a significant impact on words, and understanding words without context may lead to errors. For example, without context, the word "well" is generally considered to have a positive emotion, but in sentence 1 (shown in Figure 1), it functions as an interjection to enhance the tone. This usage does not indicate a positive meaning. Therefore, it is necessary to dynamically adjust emotions and semantics to adapt to the context.

Regarding semantics, we utilize weighted adjustments of context word embeddings with semantics.

$$E_{ds} = w_s E_c + b_s \qquad (2)$$

where $w_s$ and $b_s$ are trainable parameters, and $E_{ds} \in R^{L \times d_s}$. $d_s$ is the hidden size of the dynamic semantic vector $E_{ds}$.

Regarding emotions, we interact context word embeddings $E_c$ with emotion embeddings $E_e$ to obtain dynamic emotion vectors $E_{de}$. Emotion embeddings refer to emotion categories represented in word form, which are converted into vector embed-

dings.

$$E_{dot} = (w_c E_c + b_c) \cdot (w_e E_e + b_e)^T \qquad (3)$$
$$E_{de} = w_{ce} E_{dot} + b_{ce} \qquad (4)$$

where $w_c, b_c, w_e, b_e, w_{ce}, b_{ce}$ are trainable parameters. $E_{dot}, E_{de} \in R^{L \times d_e}$, and $d_e$ is the number of emotional categories.

We then feed the combined emotion and semantic vectors into the encoder to learn emotion-semantic representations. In this way, the model can take into account both emotion and semantics simultaneously during training to more comprehensively understand words in context.

$$V_{des} = E_{de} \oplus E_{ds} \qquad (5)$$
$$H_{des} = Enc_{des}(V_{des}) \qquad (6)$$

where $V_{des}, Enc_{des}$, and $H_{des}$ refer to the dynamic vectors, encoder, and representations for emotion-semantics, respectively. And $V_{des}, H_{des} \in R^{L \times (d_s + d_e)}$.

**Dynamic Correlation Graph Convolutional Network**. The next problem is how to focus on and learn grammatical correlations. Dependency trees clearly reflect the grammatical dependency relationships between related words (Kuncoro et al., 2016; Kiperwasser and Goldberg, 2016; Chen and Manning, 2014; Dozat and Manning, 2016). Therefore,

we use dependency trees to reflect the correlations between words. However, words with correlations in the dependency tree are not always important for understanding emotions and semantics. For example, "exciting" and "team" are more important, while "exciting" and "is" are relatively unimportant. To distinguish these correlations, we consider multiple aspects of correlation guidance, including dynamic emotion-semantic representation, part-of-speech of related words, and the dependency types between them.

We list two examples to illustrate the validity of the above correlation guidance. In sentence 2 (shown in Figure 1), the part-of-speech (ADJ and NOUN)[2] and the dependency type (CCOMP)[3] indicate that "exciting" is closely related to the key semantic "team". Conversely, the part-of-speech (ADJ and AUX)[4] and dependency type (COP)[5] indicate that the correlation between "exciting" and "is" is trivial and unimportant.

Therefore, we consider the above multiple aspects and concatenate the emotion-semantic representation, part-of-speech, and dependency type to form the guiding vector $V_{qk}$.

$$V_{qk} = H_{des} \oplus V_p \oplus V_r \tag{7}$$

where $V_p$, $V_r$ respectively denote part-of-speech embeddings , dependency type embeddings, which are randomly initialized. And $V_p, V_r \in R^{L \times L \times (d_{pr})}$, $V_{qk} \in R^{L \times L \times (d_s + d_e + 2d_{pr})}$. $d_{pr}$ is the embedding size of $V_p$ or $V_r$.

By assigning probabilities to each correlated neighbor, we aggregate the neighboring nodes that are associated in the dependency tree. Subsequently, we obtain the correlation representation $H_{cor}$.

$$p_{i,j} = \frac{a_{i,j} \cdot exp(V_{qk}[i] \cdot V_{qk}[j])}{\sum_{j=1}^{L} a_{i,j} \cdot exp(V_{qk}[i] \cdot V_{qk}[j])} \tag{8}$$

$$H_{cor} = ReLU(\sum_{j=1}^{L} p_{i,j}(W_v V_{des}[j] + b_v)) \tag{9}$$

where $p_{i,j} \in R^{L \times L \times 1}$, $H_{cor} \in R^{L \times (d_s + d_e)}$. $a_{i,j}$ is the value of the adjacency matrix about the dependency tree. When node i and node j have a direct relationship in the dependency tree, $a_{i,j}$ is 1, otherwise it is 0. $V_{qk}[i]$ and $V_{qk}[j]$ represent the

[2]ADJ: adjective, NOUN: noun
[3]CCOMP: clausal complement
[4]AUX: auxiliary
[5]COP: copula

guiding vectors of node i and node j, respectively. $W_v$ and $b_v$ are trainable parameters, and $ReLU$ is the ReLU activation function.

## 3.5 Emotion and Response Predicting

Based on the context semantics and the correlations between emotions and semantics, we predict the emotions of the conversation and generate empathetic responses.

**Emotion Predicting**. To understand the context semantics and capture important correlations, we use two aggregation networks with the same structure but different parameters to process the context semantic representation $H_{ctx}$ (Eq. 1) and correlation representation $H_{cor}$ (Eq. 9). We take the processing of the context representation as an example.

We first calculate the weights of the words in the context semantic representation $H_{ctx}$ and sum them up according to their weights to obtain the hidden layer representation $H^2$.

$$H_a^1 = Tanh(w_a^1 H_{ctx} + b_a^1) \tag{10}$$

$$P_s = Softmax(w_s^1 H_a^1 + b_s^1) \tag{11}$$

$$H^2 = \sum_{j=1}^{L} P_s[j] \cdot H_{ctx}[j] \tag{12}$$

where $H_a^1 \in R^{L \times d}$, $P_s \in R^{L \times 1}$, $H^2 \in R^d$. $w_a^1, b_a^1, w_s^1, b_s^1$ are learnable parameters, and $Tanh$ is the tanh activation function.

Then we feed the hidden layer representation into a non-linear layer to learn and generate context semantic emotion probabilities $P_{ctx}^e$.

$$H_a^2 = Tanh(w_a^2 H^2 + b_a^2) \tag{13}$$

$$P_{ctx}^e = Softmax(w_s^2 H_a^2 + b_s^2) \tag{14}$$

where $H_a^2 \in R^d$, $P_{ctx}^e \in R_e^d$, $w_a^2, b_a^2, w_s^2, b_s^2$ are learnable parameters.

Similarly, we use the same structure to construct an aggregation attention network about the correlations and obtain emotion probabilities $P_{cor}^e \in R_e^d$. We add the two types of emotion probabilities together as the overall emotion probability $P_e \in R_e^d$.

$$P_e = P_{ctx}^e + P_{cor}^e \tag{15}$$

To ensure that important information in semantics and correlations is not affected by each other, we set loss functions for them separately. We employ log-likelihood loss to optimize the parameters

during the training phase based on the emotion category and the ground truth label.

$$\mathcal{L}_{ctx}^e = -log(P_{ctx}^e(e^*)) \quad (16)$$

$$\mathcal{L}_{cor}^e = -log(P_{cor}^e(e^*)) \quad (17)$$

**Response Predicting**. Similarly, our decoder generates responses based on the context semantics and the correlations between emotion and semantics. This module takes the context semantic representation $H_{ctx}$ and the correlation representation $H_{cor}$ as input and predicts the next word at each time step t. Similar to (Li et al., 2022), we use a point generator network to capture key vocabulary in the context and correlations.

$$H = H_{ctx} \oplus H_{cor} \quad (18)$$

$$P(y_t|y < t, C) = Dec(E_{y<t}, H) \quad (19)$$

where $H \in R^{L \times (d+ds+de)}$, $P(y_t|y < t, C) \in R^V$. $V$ is the length of the vocabulary. $Dec$ represents a decoder with a pointer network.

Subsequently, we use cross-entropy as generation loss.

$$\mathcal{L}_{gen}(y_t) = -\sum_{t=1}^{T} log(P(y_t|y < t, C)) \quad (20)$$

**Total Loss**. Finally, we add the loss $\mathcal{L}_{gen}(y_t)$ and two emotion losses $\mathcal{L}_{ctx}^e$ and $\mathcal{L}_{cor}^e$ together to obtain the total loss $\mathcal{L}$. We optimize the training parameters in the model using the total loss.

$$\mathcal{L} = \mathcal{L}_{gen}(y_t) + \mathcal{L}_{ctx}^e + \mathcal{L}_{cor}^e \quad (21)$$

# 4 Experiments

## 4.1 Baselines

We compare recent state-of-the-art baselines with our model.

**Transformer** (Vaswani et al., 2017) is a vanilla Seq2Seq model, including both encoder and decoder;

**EmoPrend-1** (Rashkin et al., 2019) is a Transformer-based model that enhances empathy by incorporating emotion labels from a pre-trained emotion classifier;

**MoEL** (Lin et al., 2019) is also a Transformer-based model that softly combines various emotions with multiple decoders to generate empathetic responses;

**MIME** (Majumder et al., 2020) is a Transformer-based model, which consider polarity-based emotion clusters and emotional mimicry to generate appropriate responses;

**EmpDG** (Li et al., 2020) emphasizes the importance of user feedback and multi-resolution emotions. It uses a generative adversarial network to train the model and generate empathetic responses;

**KEMP** (Li et al., 2022) employs ConceptNet as extra knowledge to enrich the representation of implicit emotions and captures these emotions to generate appropriate responses;

**CEM** (Sabour et al., 2022) takes into account both the emotional and cognitive aspects of empathy. By incorporating reasoning knowledge, it enhances the ability to perceive and express emotions.

## 4.2 Implementation Details

We conduct experiments on the EMPATHETIC-DIALOGUES (Rashkin et al., 2019) dataset. In the dataset, the number of emotions is $d_e$=32. In the model, we use Glove (Pennington et al., 2014) as the initialization vector for word embedding, with a dimension of d=300. We set the dimension of the dynamic emotion vector to $d_s$=10. At the same time, the dimensions of the part-of-speech embedding and dependency type embedding are both set to $d_{pr}$=50. We use Biaffine Parser (Dozat and Manning, 2016) to obtain dependency relationships. For the multi-head attention networks in our model, we use a 1-layer network and set the number of heads to 2. Subsequently, we set the batch size to 16 and use the Adam optimizer (Kingma and Ba, 2014) to optimize the parameters. After training for 13500 rounds on an NVIDIA Tesla T4 GPU, the model converged.

## 4.3 Evaluation Metrics

As with previous work, we employ both automatic and manual metrics to evaluate the performance of the model.

**Automatic Evaluation Metrics**. Following (Li et al., 2022; Sabour et al., 2022), we use the following automatic metrics in our experiments: Perplexity (PPL), Accuracy (Acc), Dist-1, and Dist-2. PPL measures language fluency, which is of higher quality when the score is lower. Acc assesses the accuracy of emotion perception. Dist-1 and Dist-2 (Li et al., 2015) measure response diversity at single and double granularity, respectively.

**Human Evaluation Metrics**. Previous work scores models' responses on a scale of 1 to 5 to assess their quality (Li et al., 2020, 2022). This type of assessment is prone to inconsistent results due to differences in individual criteria (Sabour

| Models | Acc | PPL | Dist-1 | Dist-2 |
|---|---|---|---|---|
| Transformer | - | 37.73 | 0.47 | 2.04 |
| EmoPrend-1 | 33.28 | 38.30 | 0.46 | 2.08 |
| MoEL | 32.00 | 38.04 | 0.44 | 2.10 |
| MIME | 34.24 | 37.09 | 0.47 | 1.91 |
| EmpDG | 34.31 | 37.29 | 0.46 | 2.02 |
| KEMP | 39.31 | 36.89 | 0.55 | 2.29 |
| CEM | 39.11 | 36.11 | 0.66 | 2.99 |
| ESCM | **41.19** | **34.82** | **1.19** | **4.11** |

Table 1: The automatic evaluation results.

| Comparisons | Aspects | Win | Lose | $\kappa$ |
|---|---|---|---|---|
| ESCM vs. EmpDG | Emp. | **45.4** | 24.0 | 0.48 |
| | Rel. | **52.8** | 16.3 | 0.43 |
| | Flu. | **50.1** | 5.9 | 0.45 |
| ESCM vs. KEMP | Emp. | **44.0** | 20.0 | 0.57 |
| | Rel. | **53.3** | 21.0 | 0.46 |
| | Flu. | **35.4** | 13.4 | 0.41 |
| ESCM vs. CEM | Emp. | **37.3** | 19.8 | 0.58 |
| | Rel. | **48.9** | 21.5 | 0.41 |
| | Flu. | **33.8** | 11.6 | 0.47 |

Table 2: Results of human evaluation. For a more intuitive display, we remove the result of the Tie and only show Win and Lose. Where $\kappa$ is the inter-labeler agreement measured by Fleiss's kappa (Fleiss and Cohen, 1973), and $0.4 < \kappa \leq 0.6$ indicates moderate agreement.

et al., 2022). Therefore, we adopt the A/B test strategy (Lin et al., 2019; Majumder et al., 2020). Given two responses generated by the models for the same conversation, three professional crowdsourcers are required to assign ESCM a score of 1 on *Win* when the response generated by ESCM is better than the compared model. Correspondingly, when ESCM is better than or equal to the compared model, the crowdsourcers will add points for ESCM or *Tie*. Furthermore, three aspects are considered for evaluating models: empathy, relevance, and fluency. Empathy evaluates whether the responses show the right types of emotions; Relevance measures whether the reply is consistent with the theme and semantics of the context; Fluency assesses the response's readability and grammatical accuracy.

# 5 Results and Analysis

## 5.1 Main Results

**Automatic Evaluation Results**. Table 1 shows the main results of the automatic evaluation for all models. We find that early models (EmoPrend-1,

| Models | Acc | PPL | Dist-1 | Dist-2 |
|---|---|---|---|---|
| ESCM | **41.19** | 34.82 | **1.19** | **4.11** |
| w/o DESV | 39.21 | 34.10 | 1.07 | 3.52 |
| w/o DCGCN | 39.05 | **33.52** | 1.08 | 3.69 |
| w/o $V_r$ | 40.0 | 34.02 | 1.06 | 3.68 |
| w/o $V_p$ | 39.41 | 34.45 | 0.99 | 3.33 |
| w/o $V_{des}$ | 40.42 | 34.48 | 1.08 | 3.60 |

Table 3: Results of the ablation experiments.

MoEL, and MIME) are not as effective as models that focus on subtle emotions (EmpDG, KEMP, CEM). Additionally, we find that ESCM outperforms the baselines in all metrics. In terms of diversity, ESCM significantly outperforms the baselines. This suggests that focusing on the correlations between emotions and semantics helps the model capture key semantics and express informative responses. ESCM also outperforms the baselines in terms of emotion accuracy, indicating the effectiveness of the dynamic emotion-semantic vectors. Furthermore, ESCM achieves the best fluency, which indicates that the model combines emotions and semantics to express more natural language.

**Human Evaluation Results**. As shown in Table 2, ESCM outperforms the three strongest baselines in terms of empathy, relevance, and fluency. The superiority in empathy indicates that the model accurately understands and expresses emotions. The significant improvement in relevance suggests that the model captures and expresses key semantics. The superiority in fluency indicates that the model expresses more fluent responses by better understanding the context.

## 5.2 Ablation Studies

To verify the effectiveness of each component, the following experiments are conducted:

(1) **w/o DESV**: Dynamic emotion-semantic representations $H_{des}$ (in Eq. 6) are replaced by context semantic representations $H_{ctx}$ (in Eq. 1);

(2) **w/o DCGCN**: No dynamic correlation graph convolutional network (in Eqs. 7 - 9);

(3) **w/o** $V_r$/$V_p$/$V_{des}$: Without the guidance of vectors $V_r$/$V_p$/$V_{des}$ (in Eq. 7) in the dynamic correlation graph convolutional network.

The results of the ablation experiments are shown in Table 7. To verify the effectiveness of the dynamic emotion-semantic vectors, we remove $DESV$. The results show a significant decrease in emotional accuracy and diversity. The drop in

emotion accuracy suggests that dynamic emotion-semantic vectors play an important role in capturing emotions. The changes in diversity metrics demonstrate the crucial role of dynamic adjustment of emotions and semantics in precise understanding and informative expression in conversations.

We remove $DCGCN$ and its various guiding vectors to verify the effectiveness of the dynamic correlation graph convolutional network. After removing $DCGCN$, we find a significant decrease in emotional accuracy and diversity. This indicates that correlations have a significant impact on the perception of emotions and semantics, and they play an important role in expressing informative responses. We further explore the role of various guiding vectors. When part-of-speech $V_p$ or dependency types $V_r$ are removed, the emotional accuracy decreases significantly. This indicates that ESCM can aggregate effective information related to emotions based on part-of-speech or dependency types. After removing part-of-speech, dependency types, and emotion-semantic vectors respectively, the diversity decreases significantly. This indicates that these features affect the aggregation of semantic information.

Furthermore, we find that removing any module improves the fluency of responses but decreases diversity. This is mainly due to the fact that the ablation models express responses with more fluent yet less information, such as trivial sentences.

### 5.3 Correlation Analysis

To further explore the correlations, we conduct a statistical analysis of the correlations in the EMPATHETIC-DIALOGUES dataset (see appendix A for details). We extract 1138 correlations from the dialogue dataset, which are used a total of 151242 times in the dialogue. This indicates that the correlation structure is frequently and repeatedly used in the dialogue. In addition, we find that emotions are mainly expressed through three parts of speech: adjectives, nouns, and verbs. This is consistent with linguistic research (Foolen et al., 2012) on the parts of speech of emotional words. At the same time, we also find that the "preposition + noun" structure is frequently used, which is also consistent with linguistic research (Foolen et al., 2012). In each emotion type of empathetic dialogue, the frequently used correlation structures are similar, but the frequency of use may differ.

| Models | T1 (s) | T2 (s) | Num |
|--------|--------|---------|--------|
| KEMP | 0.17 | 4378.11 | 26,000 |
| CEM | 0.22 | 4438.68 | 20,000 |
| ESCM | 0.20 | 2733.94 | 13,500 |

Table 4: Results of time consumption. T1 represents the average per-iteration time, T2 stands for the convergence time, and Num is the number of iterations needed for convergence.

| Models | PPL | Acc | Dist-1 | Dist-2 | Size(G) |
|--------|-------|-------|--------|--------|---------|
| KEMP | 39.31 | 36.89 | 0.55 | 2.29 | 6.02 |
| CEM | 39.11 | 36.11 | 0.66 | 2.99 | 5.57 |
| V1 | 33.74 | 40.21 | 0.98 | 3.17 | 4.64 |
| V2 | 34.82 | 41.63 | 1.07 | 3.52 | 6.00 |
| V3 | 34.82 | 41.19 | 1.19 | 4.11 | 8.10 |

Table 5: Results under restricted resource consumption. 'Size' denotes the GPU memory required to train the model. We list three ESCM models with different $d_{pr}$ values, where V1, V2, and V3 represent models with $d_{pr} = 10$, $d_{pr} = 20$, and $d_{pr} = 50$, respectively.

### 5.4 Time and Resource Consumption

To further demonstrate the efficacy of the model, we conduct analytical experiments on time and resource consumption.

**Time Consumption**. As shown in Table 4, we calculate the time for our model and baselines to converge during training. The results show that our model does not have significantly higher average per-iteration time compared to the baselines. This is mainly because: although the intermediate variable $V_{qk}$ intuitively has higher dimensionality, in practice the dialogue context length L is short, so the impact on time is not substantial. Meanwhile, since our model can better understand the dialogue, it requires fewer iterations. Therefore, the total training time is actually significantly less than the baselines.

**Resource Consumption**. As shown in Table 5, we compute the GPU memory consumption of the models. Due to the higher dimensionality of the intermediate variable $d_{pr}$ in our model, our model requires slightly more resources than the baselines (see last row). However, even when reducing the GPU consumption of our model to be comparable to the baselines, our model still significantly outperforms the baselines overall (see penultimate row).

| Emotion | Annoyed |
|---|---|
| Conext | I hate when my little brother throws food on the table just to annoy me |
| Gold | That is a terrible waste of food, how old is your little brother? |
| CEM | I would be mad too |
| KEMP | I am sorry to hear that. I hope you can get it back soon. |
| ESCM | **That is really annoying. I hate when that happens.** |
| Emotion | Proud |
| Conext | I am constantly amazed by my sister's art skills. I am not sure where she got them ... my brother and I can not draw to save our lives. |
| Gold | What does she draw? |
| CEM | That is awesome! I bet she's a great job! |
| KEMP | I am sure you will do great! |
| ESCM | **That is really cool. I bet you are very proud of her.** |

Table 6: Case Study of ESCM and benchmarks. Words with Marked colors in the sentence are rich in emotion.

## 5.5 Case Study

As shown in Table 7, we select two strongest baselines and compare them with ESCM through sample analysis. In the first case, The speaker is annoyed due to the fact that "the younger brother threw food onto the table". The baselines do not accurately understand the emotional expression "annoyed" and the event it described. However, ESCM understands and expresses "annoying" correctly, and gives a response indicating disgust towards the event. This indicates that ESCM is able to capture key semantics through correlations. In the second case, the speaker expresses the emotion of "proud" using the word "amazed" with a surprised emotion. The baselines do not understand the emotions and semantics involved, while ESCM accurately understands the emotions and expresses an empathetic response. This demonstrates the effectiveness of building dynamic emotion-semantics.

## 6 Conclusion and Future Work

This paper proposes ESCM, which introduces two characteristics of emotions in the linguistic expression process: variability and the correlations between emotions and semantics. The proposed model constructs a dynamic emotion-semantic vector to reflect variability and uses a dependency tree-based dynamic correlation graph convolutional network to learn correlations. Both automatic and manual metrics demonstrate the effectiveness of the model. Furthermore, we conduct statistical analysis experiments. The results show that correlations are frequently used in the dialogue. Additionally, we find that the correlation structures in the dialogue are consistent with linguistic research.

To further investigate the correlation between emotion and semantics, we will take into account pre-trained knowledge, multilinguality, personalization, and other factors in future work.

## Limitations

The limitations of our work are as follows: (1) Pre-trained models have become the mainstream nowadays. To further explore the impact of pre-trained models, we constructed a pre-trained ESCM model. Since the word coverage of the EMPATHETIC-DIALOGUES dataset in the vocabulary of pre-trained models is only 51.8%, we only surpassed the baseline Emp-RFT (Kim et al., 2022) on two metrics (Acc: 42.44 (42.08), Dist-2: 9.91 (4.48)). In the future, we will further explore the impact of pre-trained models on correlations. (2) The correlations between emotions and semantics discussed in this paper are only applicable to English. However, different languages may have different types of correlations (Foolen et al., 2012). Therefore, we will investigate the correlations between emotions and semantics in multilingual contexts in the future. (3) Intuitively, there are individual differences in the expression of emotions. Due to data limitations, we did not consider this personalized factor. We will involve more research on correlations and personalization in the future.

## Ethical Considerations

The potential ethical implications of our work are as follows: (1) Dataset: EMPATHETICDIALOGUES is an open-source, publicly available dataset for empathetic response generation. In the dataset, the original provider has filtered information about personal privacy and ethical implications (Rashkin et al., 2019). (2) Models: Our baselines are also open source, and they have no permission issues. Since our model is trained on a healthy

dataset, it does not generate discriminatory, abusive, or biased responses to users.

## Acknowledgments

We would like to express our sincere gratitude to the reviewers for their diligent evaluation and constructive feedback, which helped improve the quality of this paper. In addition, we appreciate the insightful discussions and comments from the authors, which stimulated valuable thinking around this work. Their diverse perspectives and experience shared through feedback have contributed immensely to the development of this research. This work was supported by National Natural Science Foundation of China (No.61976054).

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

| Correlations/Percentage | Examples |
|---|---|
| ROOT-root-ADJ-b(7.22) | joyful, scary |
| NOUN-amod-ADJ-b(6.51) | time good |
| VERB-dep-ADJ-b(4.66) | felt proud |
| ADP-pobj-NOUN-b(3.89) | into accident |
| ADJ-nsubj-PRON-f(3.4) | nice it |

Table 7: Top 5 correlations that are frequently used in conversations.

Jeffrey Pennington, Richard Socher, and Christopher D Manning. 2014. Glove: Global vectors for word representation. In *Proceedings of the 2014 conference on empirical methods in natural language processing (EMNLP)*, pages 1532–1543.

Günter Radden. 1998. The conceptualisation of emotional causality by means of prepositional phrases. *Speaking of emotions: Conceptualisation and expression*, pages 273–294.

Hannah Rashkin, Eric Michael Smith, Margaret Li, and Y-Lan Boureau. 2019. Towards empathetic open-domain conversation models: A new benchmark and dataset. In *ACL*, page 5370–5381.

Sahand Sabour, Chujie Zheng, and Minlie Huang. 2022. Cem: Commonsense-aware empathetic response generation. In *Proceedings of the AAAI Conference on Artificial Intelligence*, Virginia, USA. AAAI Press.

Ashish Vaswani, Noam Shazeer, Niki Parmar, Jakob Uszkoreit, Llion Jones, Aidan N Gomez, Łukasz Kaiser, and Illia Polosukhin. 2017. Attention is all you need. In *NeurIPS*, page 5998–6008.

Liuping Wang, Dakuo Wang, Feng Tian, Zhenhui Peng, Xiangmin Fan, Zhan Zhang, Mo Yu, Xiaojuan Ma, and Hongan Wang. 2021. Cass: Towards building a social-support chatbot for online health community. In *PACMHCI*, volume 5, pages 1–31.

Chujie Zheng, Yong Liu, Wei Chen, Yongcai Leng, and Minlie Huang. 2021. CoMAE: A multi-factor hierarchical framework for empathetic response generation. In *Findings of the Association for Computational Linguistics: ACL-IJCNLP 2021*, pages 813–824, Online. Association for Computational Linguistics.

Peixiang Zhong, Di Wang, Pengfei Li, Chen Zhang, Hao Wang, and Chunyan Miao. 2021. Care: Commonsense-aware emotional response generation with latent concepts. In *AAAI*, volume 35, pages 14577–14585.

Peixiang Zhong, Chen Zhang, Hao Wang, Yong Liu, and Chunyan Miao. 2020. Towards persona-based empathetic conversational models. *arXiv:abs/2004.12316*.

# A    Appendix

**Frequency Phenomenon**. As shown in Figure 3, we list the correlation statistical results. (1) As

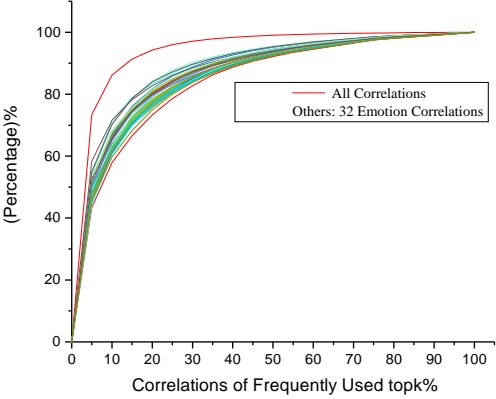

Figure 3: The percentage of frequently used correlations to the total number of correlations.

shown in "All Correlation" in Figure 3 (the highest red line), for the overall correlations in the dataset, the top 10% of frequently used correlations account for more than 80% of the total number of correlations. That is, $\frac{C_f}{C_{total}}$=86.17%, where $C_f$ is the number of frequently used correlations in the top 10%, and $C_{total}$ is the total number of correlations used in the dataset.

(2) As shown in "Others: 32 Emotion Correlations" in Figure 3, for each type of emotional empathetic dialogue, the top 20% of frequently used correlations account for 80% of the total number of correlations. That is, $\frac{C_f^e}{C_{total}^e}$ is approximately 80%. For example, the number of empathetic dialogues expressing "joyful" emotion is $C_{total}^{joyful}$=6083, and the number of frequently used associations in the top 20% is $C_f^{joyful}$=4945, so $C_f^{joyful}/C_{total}^{joyful}$ is approximately 81.13%.

**Part-of-Speech Phenomenon**. As shown in Table 7, we list examples of the most commonly used correlation structures. Taking "NOUN-amod-ADJ-b" as an example, "NOUN-amod-ADJ" represents a noun and an adjective linked together by the "amod" dependency type. "b" refers to the emotion word as the second word, and "f" indicates that the emotion word is the first word. Specifically, "ROOT" refers to the root node of the dependency tree. In the frequently used correlations, emotions are mainly expressed through three parts of speech: adjectives, nouns, and verbs, which is consistent with linguistic research (Foolen et al., 2012) on emotion words.

**Correlation Structure Phenomenon**.    As shown in Table 7, we list the most commonly used

| Type | Top1 | Top2 |
|---|---|---|
| surprised(8281) | ROOT-root-ADJ-b(6.48) | NOUN-amod-ADJ-b(5.72) |
| excited(6471) | NOUN-amod-ADJ-b(8.27) | ROOT-root-ADJ-b(7.87) |
| annoyed(4210) | ROOT-root-ADJ-b(7.01) | ROOT-root-VERB-b(5.11) |
| proud(5915) | ROOT-root-ADJ-b(11.78) | NOUN-amod-ADJ-b(5.93) |
| angry(4859) | ROOT-root-ADJ-b(10.91) | ADJ-nsubj-PRON-f(5.97) |
| sad(3945) | NOUN-amod-ADJ-b(7.25) | ROOT-root-ADJ-b(5.65) |
| grateful(6487) | ROOT-root-ADJ-b(8.89) | NOUN-amod-ADJ-b(6.98) |
| lonely(3083) | NOUN-amod-ADJ-b(11.39) | ADP-pobj-NOUN-b(6.94) |
| impressed(5483) | ROOT-root-ADJ-b(9.5) | NOUN-amod-ADJ-b(7.5) |
| afraid(5276) | ROOT-root-ADJ-b(6.99) | NOUN-amod-ADJ-b(4.78) |
| disgusted(4326) | ROOT-root-ADJ-b(7.19) | VERB-dep-ADJ-b(5.27) |
| confident(4203) | VERB-dep-ADJ-b(7.52) | ROOT-root-ADJ-b(7.47) |
| terrified(5436) | ROOT-root-ADJ-b(5.98) | NOUN-amod-ADJ-b(4.67) |
| hopeful(4635) | NOUN-amod-ADJ-b(10.68) | ROOT-root-ADJ-b(7.62) |
| anxious(4946) | ROOT-root-ADJ-b(8.07) | VERB-dep-ADJ-b(6.29) |
| disappointed(3861) | NOUN-amod-ADJ-b(7.15) | ROOT-root-ADJ-b(6.81) |
| joyful(6083) | ROOT-root-ADJ-b(8.83) | NOUN-amod-ADJ-b(5.85) |
| prepared(3617) | ROOT-root-ADJ-b(6.99) | ADP-pobj-NOUN-b(6.58) |
| guilty(4979) | VERB-dep-ADJ-b(12.07) | VERB-acomp-ADJ-b(8.44) |
| furious(4978) | ROOT-root-ADJ-b(9.62) | ADJ-nsubj-PRON-f(4.74) |
| nostalgic(4012) | NOUN-amod-ADJ-b(9.92) | ROOT-root-ADJ-b(4.99) |
| jealous(5210) | NOUN-amod-ADJ-b(7.98) | ROOT-root-ADJ-b(6.93) |
| anticipating(4502) | NOUN-amod-ADJ-b(8.93) | ROOT-root-ADJ-b(6.13) |
| embarrassed(3289) | ROOT-root-ADJ-b(5.17) | VERB-dep-ADJ-b(4.68) |
| content(6632) | ROOT-root-ADJ-b(8.05) | NOUN-amod-ADJ-b(7.86) |
| devastated(3905) | ADP-pobj-NOUN-b(6.17) | ROOT-root-ADJ-b(4.87) |
| sentimental(3123) | NOUN-amod-ADJ-b(8.65) | ROOT-root-ADJ-b(6.56) |
| caring(4041) | ROOT-root-ADJ-b(6.68) | NOUN-amod-ADJ-b(5.57) |
| trusting(4193) | NOUN-amod-ADJ-b(6.65) | ROOT-root-ADJ-b(6.39) |
| ashamed(3758) | VERB-dep-ADJ-b(9.05) | NOUN-amod-ADJ-b(4.68) |
| apprehensive(4267) | NOUN-amod-ADJ-b(8.39) | ROOT-root-ADJ-b(7.05) |
| faithful(3236) | NOUN-amod-ADJ-b(8.0) | ROOT-root-ADJ-b(5.87) |

Table 8: Top 2 frequently used correlations in conversations for each emotional dialogue. The numbers in the first column indicate the total number of correlations for the emotional dialogue, while the numbers in the other columns represent the percentage of times the correlations are used.

correlation structures for each type of emotion. We also find that the "preposition + noun" method is frequently used, which is consistent with linguistic research (Foolen et al., 2012).

As shown in Table 8, we list the most commonly used correlations in various emotional dialogues, with the numbers in parentheses indicating the probability of their usage. In each type of emotional empathetic dialogue, the frequently used correlation structures are similar, but the frequency of use may vary.