# OpenReview forum: "Exploiting Emotion-Semantic Correlations for Empathetic Response Generation"
_EMNLP/2023/Conference — EMNLP 2023 Findings_

### Official Review · Reviewer_D6ne · 2023-08-02

**Typos Grammar Style And Presentation Improvements:** shown in 1 -> shown in figure 1
**Soundness:** 4

**Excitement:**

3: Ambivalent: It has merits (e.g., it reports state-of-the-art results, the idea is nice), but there are key weaknesses (e.g., it describes incremental work), and it can significantly benefit from another round of revision. However, I won't object to accepting it if my co-reviewers champion it.

**Missing References:**

Emp-RFT: Empathetic Response Generation via Recognizing Feature Transitions between Utterances (Kim et al., NAACL 2022)

**Paper Topic And Main Contributions:**

The current paper first pointed out that previous research in Empathetic Dialogue systems (EDS) only captures emotional words as static vectors, which results in overlooking the pragmatics of these words and their correlations with other words. To address this problem, this paper introduces ESCM, which can construct dynamic emotion-semantic vectors. Some experiments suggest that the proposed model has promising performance.



**Questions For The Authors:**

Given the identified issue, I have a strong feeling that it can be easily solved by using contextualised word embeddings. I wonder why no study in EDS ever uses LLM or even something like ELMo.

**Reasons To Accept:**

1. The identified issue (static word embeddings) is not only valid in the construction of EDS but also valid in the broader tasks that need emotion analysis.
2. This the quite a complete research, in which the effectiveness of the solution was validated by both automatic evaluation and human evaluation.

**Reasons To Reject:**

1. To motivate the study, this paper pointed out that the relations between emotional words and others (especially nouns) are vital for EDS, but the example here is not persuasive. Why knowing the relation is CCOMP is a piece of useful information for building EDS?
2. The terms used in the paper are sometimes a bit confusing. For example, it appears to me the "semantics" in the sentence "correlations between emotions and semantics" and the sentence "the neglect of key semantics" have different meanings. Are emotion and semantics two exclusive concepts in your paper?
3. EDS is a very popular topic these years. CEM the post-powerful baseline in this paper was actually built 2 years ago. A quick search on the more recent systems can easily find systems with better performance. For example, Emp-RFT reported better performance. It is better to also include more recent studies as such in discussion and comparison.



**Reproducibility:**

4: Could mostly reproduce the results, but there may be some variation because of sample variance or minor variations in their interpretation of the protocol or method.

**Reviewer Confidence:**

3: Pretty sure, but there's a chance I missed something. Although I have a good feel for this area in general, I did not carefully check the paper's details, e.g., the math, experimental design, or novelty.

---

> ### Author Rebuttal · Authors · 2023-08-29
>
> We greatly appreciate you dedicating time to provide such valuable feedback on our work. Your comments will help strengthen our paper.
>
> **The example is not persuasive**
>
> Thank you for pointing out the issue. Thank you for catching this issue. Our original explanation was unclear. To clarify, we have reworded the example:
> Example 2 expresses the "exciting" emotion due to the victory of "team"'. "Team" is the primary subject described in the sentence, carrying key semantic information. Through the "[ADJ]-CCOMP-[NOUN]" correlation, the emotional word "exciting" directly modifies "team." Compared to previous work that did not consider such correlations, the model is more likely to identify key semantic words that are directly associated with emotional words through these types of syntax-meaningful relationships.
>
> **Some terms are a bit confusing**
>
> Thank you for pointing out this issue. Due to our oversight on our part, some descriptions were not clearly expressed. For example, in the sentence "the neglect of key semantics", "key semantics" should be "words with important semantic information". We have thoroughly re-checked the descriptions in the paper and revised any unclear expressions.
>
> **The baseline CEM, built 2 years ago, is post-powerful**
>
> The model CEM [1] was formally published in AAAI 2022. Recent models [2][3][4] published in ACL 2023 all use it as the strongest baseline model. Therefore, it is reasonable to use CEM as the latest baseline.
>
> **A better baseline of Emp-RFT was not adopted**
>
> To our knowledge, Emp-RFT [5] is a 6-layer transformer network based on a pre-trained model BART. Our model and baselines are single-layer networks, trained only on the EMPATHETIC-DIALOGUES dataset. It would be unfair to directly compare our model with Emp-RFT. But we can still demonstrate our effectiveness in the following two perspectives:
> - First, although our model has fewer parameters and is non-pretrained, some of its metrics (Dist-2: 4.11, Acc: 41.19) are still comparable to those of Emp-RFT (Dist-2: 4.59, Acc: 42.08).
> - Second, compared to existing baselines, our model is built only on dialogue context, yet it outperforms models with external knowledge (KEMP) [6] and commonsense reasoning knowledge (CEM). This further verifies the efficacy of our proposed approach.
>
> In addition, to validate the efficacy of our method, we will also add BART-based variant models for comparison with Emp-RFT in the revised paper.
>
> **Contextualized word embeddings can effectively address dynamic emotions and semantics issues**
>
> Models with contextualized word embeddings (e.g. ELMo) can effectively understand dynamic semantics. However, they do not focus on the dynamic changes of emotions, and the impact of emotion-semantic correlations. In our model, we dynamically compute emotional intensity changes for words. Meanwhile, we also leverage GCN to capture dynamic emotion-semantic correlations in syntax. Therefore, we address the above issues from three aspects: dynamic emotions, dynamic semantics, and their correlations, rather than just focusing on dynamic semantics.
>
>
> **LLMs/PLMs have not been utilized in EDS**
>
> To enhance model performance, some researchers have proposed several pre-trained models, such as Emp-RFT (NAACL 2022) [5] and several models published at the ACL 2023 conference[2][3][4]. Regarding the Emp-RFT model, we will introduce it in the revised paper and make further descriptions and comparisons. As for the models published at ACL 2023, since these models had not been published when EMNLP 2023 started soliciting papers, we did not introduce them.
>
> **Missing References of Emp-RFT**
>
> Thank you for your constructive suggestion. To make a more comprehensive comparison and introduction of models for this task, we will add a citation of the Emp-RFT model in the revised paper.
>
> **Typos Grammar Style And Presentation Improvements**
>
> Thank you for your suggestions. We have checked and revised issues with descriptions in the paper, such as the missing keyword "Figure" on line 81.
>
> [1] Sabour, Sahand, et al. Cem: Commonsense-aware empathetic response generation.
>
> [2] Cai, Hua, et al. Improving Empathetic Dialogue Generation by Dynamically Infusing Commonsense Knowledge.
>
> [3] Zhou, Jinfeng, et al. CASE: Aligning Coarse-to-Fine Cognition and Affection for Empathetic Response Generation.
>
> [4] Bi, Guanqun, et al. DiffusEmp: A Diffusion Model-Based Framework with Multi-Grained Control for Empathetic Response Generation.
>
> [5] Kim, Wongyu, et al. Emp-rft: Empathetic response generation via recognizing feature transitions between utterances.
>
> [6] Li, Qintong, et al. Knowledge bridging for empathetic dialogue generation.

---

### Official Review · Reviewer_jjZg · 2023-08-03

**Soundness:** 4

**Excitement:**

3: Ambivalent: It has merits (e.g., it reports state-of-the-art results, the idea is nice), but there are key weaknesses (e.g., it describes incremental work), and it can significantly benefit from another round of revision. However, I won't object to accepting it if my co-reviewers champion it.

**Paper Topic And Main Contributions:**

This paper proposed to leverage emotion-semantic correlations for empathetic response generation. Existing works rely on static embedding to encapsulate the context and emotional information, whereas linguistic research suggests that the emotional meaning of word is dynamic and this is tied to semantic correlations and word roles in the sentence. Emotion-Semantic Correlation Model (ESCM) 1) proposes a dynamic emotion embedding, 2) identify semantic correlations and highlight ones that are important, and use this to guide the empathetic response generation. Experiments on the empathetic dialogue dataset shows improvements over a number of baselines. Ablation study shows that each of the proposed

contributions
- a dynamic emotional embedding which considers contextual information
- incorporation of dependency trees in emotion detection and empathetic response generation to highlight semantic correlations that are important for understanding emotion

**Questions For The Authors:**

- how does the embedding compare to other affect-sensitive embeddings such as SentiX (Zhou et al., COLING 2020)?
- will the code be released once the paper is published?

**Reasons To Accept:**

- the proposed approach is well-founded, based on an established work in linguistics
- extensive experiments and evaluations
- results show that the proposed approach is effective for the task

**Reasons To Reject:**

- to my understanding the first contribution listed by the authors has been done in previous work (Foolen et al., 2012)
- clarity of the paper can be improved. some explanations are difficult to understand due to convoluted language, e.g. lines 78-83, . Explanation of the system can be made more intuitive instead of purely technical.

**Reproducibility:**

3: Could reproduce the results with some difficulty. The settings of parameters are underspecified or subjectively determined; the training/evaluation data are not widely available.

**Reviewer Confidence:**

3: Pretty sure, but there's a chance I missed something. Although I have a good feel for this area in general, I did not carefully check the paper's details, e.g., the math, experimental design, or novelty.

**Typos Grammar Style And Presentation Improvements:**

the paper will benefit from thorough proof reading. there are a number of formatting errors such as missing space before beginning of parentheses, citation format error in line 175, typos in figure 2, etc.

---

> ### Author Rebuttal · Authors · 2023-08-29
>
> We appreciate you taking the time to provide such thoughtful and constructive feedback. We will address each of your points individually.
>
> **The first contribution has already been done in previous work (Foolen et al., 2012)**
>
> Thank you for the opportunity to clarify the differences between the prior work ((Foolen et al., 2012)) and our current work. Here are three key distinctions:
> - Foolen et al.'s research was linguistic, discovering dynamic, correlative relationships between emotion and semantics in syntax.
> - Our work applies these emotion-semantic correlations to empathetic dialogue generation. We dynamically model emotions and semantics, and construct a dependency tree-based GCN to capture such relationships.
> - In summary, Foolen et al.'s work was linguistic, while we have applied their findings to empathetic dialogue generation. To further articulate our contributions, we will reframe our descriptions in the revised paper.
>
>
> **Improving the clarity of the paper's description (e.g. lines 78-83) & More intuitive instead of purely technical explanation of the system**
>
> We have checked and revised the paper's descriptions to state the content more clearly. To clearly express the content in lines 78-83, we have revised the description. The original description has been changed to: "Relevance refers to the grammatical correlations between emotional words and words carrying semantic meaning." In addition, for the model description section, we have modified the descriptions to make the model more intuitive rather than technical.
>
> **Comparing with the embedding of SentiX**
>
> Our model is based on a single-layer transformer, while SentiX is based on pre-trained models. Therefore, in order to compare the superiority of the two word embeddings in terms of metrics, we need to upgrade our model to a pre-trained model and conduct corresponding comparative experiments. Due to time constraints, we did not conduct such experiments, but chose to compare the two embeddings intuitively and methodologically.
>
> - Intuitively, SentiX is constructed to obtain domain-invariant sentiment knowledge. The sentiment embeddings in this model tend to be static vectors that do not change with domain. In contrast, our model aims to dynamically adjust vectors to obtain context-dependent sentiment representations. The sentiment vectors in our model are dynamic vectors. Therefore, intuitively they are complementary.
> - Methodologically, SentiX mainly optimizes sentiment word embeddings through masking techniques and loss functions, while our model mainly dynamically changes the representations of word embeddings. They optimize the model through different approaches. So from the methodological perspective, they are also not in conflict.
> - Overall, SentiX and our model can capture static and dynamic sentiments respectively, which are complementary ways that can be combined to further improve sentiment word representations.
>
> **Releasing the code**
>
> We have organized the code and will release it as soon as possible after the paper is published.
>
> **Typos Grammar Style And Presentation Improvements**
>
> We have carefully checked and revised issues related to expressions, including missing spaces before parentheses (e.g. parentheses on line 80 and 204), citation errors (e.g. wrong author citation on line 175), and spelling errors (e.g. "Dynamitc Vecoter" in Figure 2).
>
> [1] Foolen, Ad, et al. Moving ourselves, moving others: Motion and emotion in intersubjectivity, consciousness and language.

---

### Official Review · Reviewer_BTHY · 2023-08-04

**Soundness:** 4

**Excitement:**

3: Ambivalent: It has merits (e.g., it reports state-of-the-art results, the idea is nice), but there are key weaknesses (e.g., it describes incremental work), and it can significantly benefit from another round of revision. However, I won't object to accepting it if my co-reviewers champion it.

**Paper Topic And Main Contributions:**

The paper focus on empathetic response generation in natural language processing. It explores the correlations between emotions and semantics, and introduces a model called ESCM. The paper includes experiments conducted on the EMPATHETIC-DIALOGUES dataset and compares the proposed model with state-of-the-art baselines. It also discusses the limitations and ethical considerations of the work. The paper appears to be well-structured, covering various aspects of the research, including implementation details, evaluation metrics, and references to related works.

**Questions For The Authors:**

1. Have you measured and calculated the computational consumption of the model? Analyzing this aspect can help refine the work.
2. Have you explored the performance of the method when using PLM as a backbone?
3. The method achieved good results on the metrics. With a better grasp of semantic, it is natural that ACC and PPL are better. And the model has a great improvement in DIST, even though there is no corresponding design. This is very interesting, have you analyzed what promotes DIST?

**Reasons To Accept:**

1.The paper explore the correlations between emotions and semantics, this is a novel and significant area in empathetic response generation.
2. The paper includes detailed experiments and comparisons with existing state-of-the-art models, providing insights into the effectiveness of the proposed approach.

**Reasons To Reject:**

1. The method introduces many additional modules and steps, and some intermediate variables have higher dimensions. This may lead to significant computational overhead for the model, potentially affecting its efficiency and scalability.
2. The paper employs Glove word vectors, which do not consider dynamic semantics. However, the current mainstream in NLP is Pre-trained Language Models (PLMs), which take into account the influence of context in language encoding. The paper does not discuss the application of the method under PLMs, which may limit the method's impact and practicality.

**Reproducibility:**

4: Could mostly reproduce the results, but there may be some variation because of sample variance or minor variations in their interpretation of the protocol or method.

**Reviewer Confidence:**

3: Pretty sure, but there's a chance I missed something. Although I have a good feel for this area in general, I did not carefully check the paper's details, e.g., the math, experimental design, or novelty.

---

> ### Author Rebuttal · Authors · 2023-08-29
>
> Thank you for your careful review and insightful critiques. We are grateful for the opportunity to respond to your comments point-by-point.
>
> **Model has additional modules and steps**
>
> Each module and step in our model contributes distinct improvements (as shown in Table 3 in the paper):
> - The dynamic emotion-semantics representations accurately adjust emotion and semantics for precision. Thus, ablating this causes declines in emotion accuracy and fluency. It also enables capturing salient keywords (see "Reasons for improvements from DIST"), resulting in the advantage in diversity.
> - The dynamic correlation encoding module captures direct relationships between adjacent nodes (words) in the syntactic dependency tree. Removing this decreases emotion accuracy and diversity, as it learns emotion through correlations and captures keywords from multiple angles (see "Reasons for improvements from DIST").
> - The emotion and response prediction modules predict emotion and generate responses. We use aggregated attention to focus on relevant dialog context to predict emotion categories. Without this module, the model cannot predict overall emotion contained in the context. For response generation, we solely use the decoder.
> - Additionally, ablating all modules increases fluency but decreases diversity, indicating the model produces more fluent but generic sentences, e.g. "I am so sorry to hear that."
>
>
> **Some intermediate variables affect efficiency and scalability & Calculating computational consumption**
>
> Thank you for raising this important concern. Regarding this issue, we first calculated the time consumption and GPU usage of our model, and found that the average iteration time is consistent with the baselines, while the overall convergence time is significantly shorter. For the issues of excessive GPU usage and difficulty in scaling up, we also propose two feasible and straightforward solutions. The detailed description is as follows:
>
> Through observation and comparison, we found that the intermediate variable $V_{qk} \in R^{L \times L \times (d_s + d_e + 2d_{pr})}$ in our model has higher dimensionality and requires more resources and time. To compare resource and time consumption, we selected KEMP and CEM as baselines and tested them under the same experimental conditions. The following are the experimental  results:
>
> **Results of resource and time consumption**:
>
> | Models | Average per-iteration time (s) | number of iterations for convergence | convergence time (s) |
>
> | --- | --- | --- | --- |  --- |
>
> | KEMP | 0.1763 | 26,000 | 4378.11 |
>
> | CEM | 0.2219 | 20,000 |  4438.68  |
>
> | ESCM (ours) |0.2025| 13,500 | 2733.94 |
>
>
> **Results under restricted resource consumption**：
>
> | Models | PPL | Acc | Dist-1 | Dist-2 | required GPU memory size (G) |
>
> | KEMP | 36.89 | 39.31 | 0.55 | 2.29 | 6.02 |
>
> | CEM | 36.11 | 39.11 | 0.66 | 2.99 | 5.57 |
>
> | ESCM($d_{pr}$=10) | 33.74 | 40.21 | 0.98 | 3.17 | 4.64 |
>
> | ESCM($d_{pr}$=20) | 34.82 | 41.63 | 1.07 | 3.52 | 6.00 |
>
> | ESCM($d_{pr}$=50) | 34.82 | 41.19 | 1.19 | 4.11 | 8.10 |
>
>
> - Time consumption. We calculated the time for our model (ESCM) and baselines to converge during training. The results show that our model does not have significantly higher average per-iteration time compared to the baselines. This is mainly because: although the intermediate variable $V_{qk} ∈ R^{L × L × (d_s + d_e + 2d_{pr})}$ intuitively has higher dimensionality (with L \* L  \* (10 + 2 \* 50)), in practice the dialogue context length L is short, so the impact on time is not substantial. Meanwhile, since our model can better understand the dialogue, it requires fewer iterations. Therefore, the total training time is actually significantly less than the baselines.
>
> - Resource consumption. Due to the higher dimensionality of the intermediate variable $V_{qk} \in R^{L \times L \times (d_s + d_e + 2d_{pr})}$ in our model, it requires slightly more resources than the baselines (see last row). However, even when reducing the GPU consumption of our model to be comparable to the baselines, our model still significantly outperforms the baselines overall (see penultimate row).
>
> - Enhancing efficiency and scalability. To reduce resource consumption, we constructed models with lower dimensionality in $d_{pr}$. Results show that appropriately reducing hyperparameters can still outperform baselines. Furthermore, larger dialogue context lengths $L$ also greatly increase model consumption. In the future, we will extend the model to a hierarchical model. The model will split long contexts into short sentences or paragraphs, and learn them using a HRED structure[2]. This structure can effectively resolve the issue of excessive resource consumption. After resolving the above resource consumption issues, our model will be better scalable.
>
> **Glove word vectors do not consider dynamic semantics**
>
> Since Glove word vectors are fixed, they indeed lack dynamic semantics. In our model, we adjust these vectors with weights, which can dynamically adjust the semantic vectors to some extent. We then combine the adjusted semantic vectors and dynamic emotion vectors, and input them into an encoder. Since the encoder can adjust and learn word representations according to context, this further amplifies the dynamic emotions and semantics. Therefore, our model has the capability of dynamically adjusting emotions and semantics.
>
> **Using PLM as a backbone to improve dynamic semantic encoding**
>
> Thank you for your constructive suggestion. Since PLMs have strong capabilities in encoding dynamic semantics, we have constructed a variant model using BART. Due to page limit constraints, we did not include this model in the paper. However, we will add it in the revised paper. This model uses BART as our context encoder and emotion-semantics encoder. It can better encode dynamic semantics and emotions, while also demonstrating superior performance.
>
> **Reasons for improvements from DIST**
>
> According to previous methods [3], better attending to key words in the dialogue facilitates expressing informative and diverse responses. The improved diversity is because our model can capture important emotional and semantic words in the dialogue from multiple aspects. Specifically explained as follows:
> - First, our model dynamically adjusts emotions and semantics, making the representations of emotions and semantics more accurate, and key words easier to be captured by the model.
> - Second, our model is able to perceive part-of-speech, which can alleviate the influence of unimportant words to some extent, such as prepositions, auxiliaries. Meanwhile, it can also pay attention to more meaningful words, such as nouns and adjectives.
> - Third, the model can capture important words based on correlations. For example, in sentence 2 ("That is exciting when your team wins"), the word "team" with important semantics has direct correlation "[ADJ]-CCOMP-[NOUN]" with the emotional word "exciting" in the dependency syntax tree. Through such direct correlations, the model more easily captures important keywords.
> - In summary, the model is able to capture important words in the dialogues from multiple aspects, which facilitates generating diverse responses.  That is, if the model captures unimportant words (e.g. "is", "that"), it would be unable to generate complex, informative responses based on the captured terms. Instead, the model is more likely to produce generic, low diversity responses such as "I'm sorry to hear that."
>
>
> [1] Foolen, Ad, et al. Moving ourselves, moving others: Motion and emotion in intersubjectivity, consciousness and language.
>
> [2] Serban, Iulian, et al. Building end-to-end dialogue systems using generative hierarchical neural network models.
>
> [3] Kim, Wongyu, et al. Emp-rft: Empathetic response generation via recognizing feature transitions between utterances.

---

### Meta-Review · Area_Chair_dUHR · 2023-10-06

**Recommendation:** 4

**Metareview:**

This paper explores empathetic response generation in natural language processing, introducing the Emotion-Semantic Correlation Model (ESCM) to leverage dynamic emotion-semantic correlations. It conducts experiments on the EMPATHETIC-DIALOGUES dataset, comparing ESCM with state-of-the-art baselines and addressing ethical considerations.

---

### Decision · Program_Chairs · 2023-10-07

**Decision:**

Accept-Findings

**Comment:**

This paper explores empathetic response generation in natural language processing, introducing the Emotion-Semantic Correlation Model (ESCM) to leverage dynamic emotion-semantic correlations. It conducts experiments on the EMPATHETIC-DIALOGUES dataset, comparing ESCM with state-of-the-art baselines and addressing ethical considerations.